# NET Formation Was Reduced via Exposure to Extremely Low-Frequency Pulsed Electromagnetic Fields

**DOI:** 10.3390/ijms241914629

**Published:** 2023-09-27

**Authors:** Caren Linnemann, Filiz Sahin, Yangmengfan Chen, Karsten Falldorf, Michael Ronniger, Tina Histing, Andreas K. Nussler, Sabrina Ehnert

**Affiliations:** 1Siegfried Weller Institute for Trauma Research, BG Unfallklinik Tübingen, Eberhard Karls Universität Tuebingen, Schnarrenbergstraße 95, 72076 Tuebingen, Germany; caren.linnemann@med.uni-tuebingen.de (C.L.); andreas.nuessler@gmail.com (A.K.N.); 2Sachtleben GmbH, Haus Spectrum am UKE, Martinistraße 64, 20251 Hamburg, Germany

**Keywords:** EMF, neutrophils, neutrophil extracellular traps, fracture healing

## Abstract

Fracture-healing is a highly complex and timely orchestrated process. Non-healing fractures are still a major clinical problem and treatment remains difficult. A 16 Hz extremely low-frequency pulsed electromagnetic field (ELF-PEMF) was identified as non-invasive adjunct therapy supporting bone-healing by inducing reactive oxygen species (ROS) and Ca^2+^-influx. However, ROS and Ca^2+^-influx may stimulate neutrophils, the first cells arriving at the wounded site, to excessively form neutrophil extracellular traps (NETs), which negatively affects the healing process. Thus, this study aimed to evaluate the effect of this 16 Hz ELF-PEMF on NET formation. Neutrophils were isolated from healthy volunteers and exposed to different NET-stimuli and the 16 Hz ELF-PEMF. NETs were quantified using Sytox Green Assay and immunofluorescence, Ca^2+^-influx and ROS with fluorescence probes. In contrast to mesenchymal cells, ELF-PEMF exposure did not induce ROS and Ca^2+^-influx in neutrophils. ELF-PEMF exposure did not result in basal or enhanced PMA-induced NET formation but did reduce the amount of DNA released. Similarly, NET formation induced by LPS and H_2_O_2_ was reduced through exposure to ELF-PEMF. As ELF-PEMF exposure did not induce NET release or negatively affect neutrophils, the ELF-PEMF exposure can be started immediately after fracture treatment.

## 1. Introduction

Non-healing wounds and fractures remain a major clinical challenge. Despite the continuous development of new surgical techniques and the materials and machines that support them, 2–3% of all wounds and 5–10% of all bone fractures still do not heal properly [1]. Annual costs for the treatment of chronic wounds and fracture non-unions exceed by far a billion euros (EUR) in Germany [2]. Their treatment remains a challenge, which often includes several revision surgeries until healing can be achieved.

Considering non-unions, several efforts have been made to overcome the impaired healing of bone. As non-invasive techniques to support healing processes, pulsed electromagnetic fields were introduced, which showed positive effects on bone in different settings, e.g., fractures, osteotomies, non-unions, spinal fusion, or osteoporosis [3] (for review see [4,5]). Despite the observed positive effects on bone, exposure to ELF-PEMFs is generally not among the guidelines for treating bone defects. One reason might be the large variety of the ELF-PEMF characteristics and treatment regimens in these reports.

Electromagnetic fields in different frequency ranges can be found throughout the environment (e.g., from cell phones or microwaves) [6]. They can affect many cellular processes such as cell signaling, growth, or migration in different tissue systems (reviewed in [6]). Here, ELF-EMF can be found in the lower range of the frequency spectrum and is thought to not have any thermal effect on biological tissues [7]. Instead, the ELF-EMF generates small electromagnetic stimuli which resemble external mechanical stimulation [4]. For osteoblasts, pulsed patterns (ELF-PEMF) were shown to be favorable compared to other patterns [8].

The present study focuses on an ELF-PEMF with a fundamental frequency of 16 Hz, which is close to the EMF (~15 Hz) measured in human bones during walking [4]. This specific 16 Hz ELF-PEMF has been shown to support bone-healing in elderly patients receiving a high tibia osteotomy in a clinical study [3]. Potential mechanisms were investigated in vitro in isolated mesenchymal stem cells and osteoblasts. Exposure to the 16 Hz ELF-PEMF was reported to induce the formation of reactive oxygen species (ROS), which in turn stimulated the cells’ anti-oxidative defense mechanisms [9] protecting the cells from exposure to cigarette smoke extract [10]. Exposure to the 16 Hz ELF-PEMF improved the cell’s response to mechanical stimuli by protecting the primary cilia structure [10] and inducing the expression of PIEZO1, a mechanosensitive Ca^2+^-channel [11]. Thus, this field was chosen for further investigation.

However, the underlying mechanisms leading to the observed effects have only been studied in osteoprogenitor cells, and to date it is not known how ELF-PEMF exposure affects other cell types involved in fracture-healing. Directly after a fracture, the immune system is one of the major players, where dysregulation can also lead to severe problems in the whole healing process [12]. Here, neutrophils are the first immune cells arriving at the fracture site and play an important role in the first hours [13]. There, they can orchestrate the healing progress by releasing cytokines and chemokines, ROS, antimicrobial peptides, and neutrophil extracellular traps (NETs) [14]. NETs are composed of DNA covered with histones and antimicrobial proteins. They aid in clearing the wound/fracture site from invading pathogens. Therefore, it is not surprising that NET markers were found to be elevated in patients’ blood after trauma [15]. However, if NET formation is excessively activated, their accumulation may hinder the healing process [16]. For example, strong negative correlations have been described between the healing of diabetic foot ulcers and NET formation [17], as well as circulating NET markers [18].

NETosis, being a very rapid but unspecific defense mechanism, is activated by many stimuli in the human body. Two very common NET-stimuli include ROS and Ca^2+^-influx [19], both reported to be induced by exposure to the 16 Hz ELF-PEMF in osteoprogenitor cells [9,11]. While 16 Hz ELF-PEMF-induced ROS and Ca^2+^-influx stimulate the maturation of osteoprogenitor cells, they might activate neutrophils to excessively form NETs and thus interfere with wound-healing. In line with this assumption, the exposure of neutrophils to LF-EMFs with higher frequencies (320, 730, 880, and 2600 Hz at a magnetic field intensity of 300 µT) increased NET formation in an NADPH oxidase-dependent manner [20]. In an in vivo approach, 30 min exposure of the whole body to an LF-EMF reduced the granularity of neutrophils, indicating an activation of the cells by the LF-EMF [21].

For other immune cells, in this case, peripheral blood mononuclear cells (PBMC), the 16 Hz ELF-PEMF investigated here had no effect—but when the fundamental frequency was increased, the ELF-PEMF was able to modulate the macrophage phenotype [22]. This is in line with reports showing increased phagocytic activity, IL-1β release, and ROS production on macrophages exposed to a 50 Hz ELF-EMF [23,24]. The effects of electromagnetic fields on diverse cell types make it likely that neutrophils are also regulated when ELF-PEMFs are applied in the context of fracture-healing. Thus, this study aimed to investigate whether the 16 Hz ELF-PEMF exposure, which could previously support bone-healing [3], affects neutrophils, more specifically ROS production, Ca^2+^ flux, and as a result, NET formation. This is of special importance for the timing of clinical applications; whether it is safe to use ELF-PEMF exposure immediately after fracture treatment when different types of immune cells actively modulate wound- and fracture-healing.

## 2. Results

### 2.1. 16 Hz ELF-PEMF Exposure Does Not Affect ROS Formation or Ca^2+^ Influx

First, we examined important mechanisms in neutrophils that are known to be modulated by ELF-PEMF [11,20]. The formation of reactive oxygen species (ROS) and Ca^2+^-influx were determined because both may lead to NETosis [25,26]. Exposure to 16 Hz ELF-PEMF alone did not induce ROS formation (Figure 1A). Stimulation with phorbol myristate acetate (PMA) showed a robust increase in ROS levels, which was not affected by 16 Hz ELF-PEMF exposure. Lipopolysaccharide (LPS) also significantly induced ROS, although to a lesser extent than phorbol myristate acetate (PMA). Exposure to the 16 Hz ELF-PEMF did not change LPS-induced ROS formation. Similarly, Ca^2+^-influx was not influenced by the 16 Hz ELF-PEMF exposure itself (Figure 1B), neither in the presence nor in the absence of LPS. Only with PMA stimulation, did the exposure to the 16 Hz ELF-PEMF show a trend towards a reduction in Ca^2+^-influx. The inhibition of Piezo1 did not change the Ca^2+^-influx (Figure 1D and Appendix A), although neutrophils robustly expressed *Piezo1* (Figure 1C).

### 2.2. 16 Hz ELF-PEMF Exposure Does Not Induce NET Formation

Next, to identify the safety of ELF-PEMF exposure on neutrophils, basal NET formation was analyzed. Freshly isolated neutrophils were exposed to 16 Hz ELF-PEMF for 7 min before a Sytox Green assay was performed. Figure 2A shows the calculation of the different analysis parameters from the time course of the Sytox Green assay. The total amount of DNA released from neutrophils did not change upon exposure to 16 Hz ELF-PEMF (Figure 2B). To investigate the effect of 16 Hz ELF-PEMF on stimulated NET formation, neutrophils were activated with PMA following the ELF-PEMF exposure. PMA-stimulated neutrophils had significantly increased NET release compared to untreated control cells. Interestingly, pre-exposure to 16 Hz ELF-PEMF significantly decreased the PMA-dependent DNA release (Figure 2C). Neutrophil activation (reaction speed) was not delayed (Figure 2D), but the maximum amount of DNA released and the NET release rate (reactivity) were significantly reduced after exposure to the 16 Hz ELF-PEMF compared to PMA-stimulation alone (Figure 2E,F).

The results from the Sytox Green assay were verified via immunofluorescence staining for MPO and DNA. Exposure to 16 Hz ELF-PEMF before PMA stimulation showed a trend for a reduced number of NETosed cells compared to PMA alone (Figure 3A) and less pronounced NET formation in 16 Hz ELF-PEMF-exposed cells (Figure 3B). As seen before in the Sytox Green assay, exposure to 16 Hz ELF-PEMF alone did not induce NET formation (Figure 3A). In addition, exposure to ELF-PEMF with other fundamental frequencies did not activate NET formation (Appendix A). Similarly to the 16 Hz field, some frequencies (23.8 Hz and 26 Hz) reduced NET formation significantly.

### 2.3. 16 Hz ELF-PEMF Exposure Does Not Change MAPK Activation

Another important pathway involved in NET release is the mitogen-activated protein kinase (MAPK) pathway. This pathway was also associated with electromagnetic field exposure in different settings [27,28]. To ensure that the 16 Hz ELF-PEMF does not activate neutrophils, we analyzed the MAPK activation after exposure and stimulation with PMA and LPS. Our results show that ELF-PEMF exposure did not affect the MAPK/ERK pathway (Figure 4A). PMA induced p-ERK, but no differences between PMA alone and PMA with additional ELF-PEMF exposure could be observed. LPS stimulation led to a trend in the increased activation (phosphorylation) of p38 but not ERK, underlining different pathways used for PMA and LPS. P-Akt and cit-H3, two more major players in NET formation [29,30], were not altered by exposure to the 16 Hz ELF-PEMF or at all by any of the treatments. The levels of cit-H3 were at an overall low.

### 2.4. The 16 Hz ELF-PEMF Exposure Reduces NET Formation by Not Only PMA but Also by LPS and H_2_O_2_

NETosis can be induced through various stimulants, e.g., calcium ionophore (CI), LPS, and H_2_O_2_, all addressing different neutrophil activation mechanisms [31]. Similarly to the experimental setup with PMA, neutrophils were exposed to 16 Hz ELF-PEMF for 7 min prior to stimulations, and DNA release was quantified using the Sytox Green assay. The 16 Hz ELF-PEMF exposure did not affect the total NET release, activation time, and peak DNA release of the cells upon CI treatment (Figure 5, top row). With LPS stimulation, 16 Hz ELF-PEMF exposure significantly delayed the DNA release and decreased the total amount of extracellular DNA (Figure 5, middle row) but did not change the peak DNA release. Following H_2_O_2_ stimulation, the total amount as well as the peak DNA release was significantly reduced through 16 Hz ELF-PEMF exposure (Figure 5, bottom row) and the reaction speed remained unaffected.

## 3. Discussion

Bone fracture-healing is a well-regulated and complex process that involves various pathways and cell types. However, complications such as non-unions or delayed unions [1] have led to a need for non-invasive and adjuvant treatments. Exposure to ELF-PEMF is used to support bone fracture-healing by mimicking external physical stimuli [32]. Although electromagnetic stimulation systems have been used as an adjunct therapy to support fracture-healing for almost 40 years [33], the underlying molecular mechanisms are not yet fully understood. One of the crucial players in bone fracture-healing is neutrophils. Thus, the present project aimed to show the effect of ELF-PEMF exposure on neutrophils and examine the safety of the ELF-PEMF system regarding NETosis. One of the reasons for choosing 16 Hz frequency ELF-PEMF exposure was because it is similar to the frequency produced in the bones while walking [34], therefore mimicking the natural physical stimuli of bones. Furthermore, previous work showed that 16 Hz ELF-PEMF can successfully be used to improve fracture-healing [35].

Our study suggests that 16 Hz and 7 min of exposure to ELF-PEMF do not activate neutrophils and do not induce basal NET formation either after additional exposure to PMA or to LPS. Indeed, we even see a reduction in NET formation as demonstrated by the reduced release of total DNA and peak DNA. These results are in contrast with a previous study [20], where frequencies between 320–2600 Hz resulted in NET formation induction. This could be partially explained by the fact that higher exposure frequency resulted in increased the activation of the NADPH oxidase pathway and ROS release [20]. It is noteworthy that cell exposure at 16 Hz in our setup did not lead to an increase in ROS. This suggests that ELF-PEMF exposure conditions are important for cell responses, and conditions similar to natural stimuli might be safer and more beneficial for clinical use.

ROS formation is known to be very crucial for NET release [36] and ROS was induced by ELF-PEMF in other cell types such as osteoprogenitor cells via increased mitochondrial activity and the induction of O^2−^ and H_2_O_2_ [9]. Both bone cells and neutrophils produce ROS through NADPH (nicotinamide adenine dinucleotide phosphate) oxidases (NOX) [37], where NOX2 is generally the main actor in neutrophils [38]. In osteoclasts, the induction of ROS formation is mediated by RANK (receptor activator of nuclear factor κ B), which activates NF-κB signaling and MAPKs in a NOX2-dependent manner [39,40]. MAPKs are also involved in ROS formation and NET formation in neutrophils [41].

Interestingly, the exposure of neutrophils to 16 Hz ELF-PEMF did not induce ROS formation, either alone or with additional stimulation from PMA or LPS, and increased MAPK activation was also not observed. For safety reasons, this is very important, as ROS formation can have unwanted outcomes in neutrophils and in the surrounding tissue. In summary, upon the activation of the NADH oxidase complex through external stimuli such as bacteria or chemicals like PMA, ROS is generated. This activation is mainly orchestrated by the MAPK pathway [42]. The formation of ROS enables the release of MPO and neutrophil elastase (NE), which makes the release of NETs possible by breaking the nuclear envelope [43]. Meanwhile, ROS also facilitates peptidyl-arginine deaminase 4 (PAD4)-induced chromatin decondensation and histone citrullination [44]. Azzouz et al. reported that excessive ROS formation induces DNA damage and affects DNA repair pathway, helping with chromatin decondensation during NET formation [19]. Hence, the release of ROS molecules triggers NET release. However, no increased ROS formation in neutrophils exposed to ELF-PEMF could be observed in our study. In other immune cells (monocytes), ROS was induced through ELF-PEMF exposure and this induction improved anti-bacterial defense capabilities [23,24]. Thus, a strong reduction in ROS would also be undesirable in neutrophils.

Similarly to the increased ROS formation seen in previous studies, there is evidence that ELF-PEMF exposure acts on Ca^2+^ influx and signaling. Petecchia et al. showed that exposure to the PEMF of bone marrow stroma-derived human mesenchymal cells promotes osteogenic differentiation via an increase in cytosolic Ca^2+^, especially in the early stages [10]. In addition, osteoprogenitor cells also showed increased Ca^2+^-influx upon 16 Hz ELF-PEMF treatment, which promotes osteogenesis [11]. In neutrophils, the ELF-PEMF exposure at the calcium ion cyclotron resonance frequency (7 Hz) resulted in increased ROS production, suggesting that specific PEMF exposures can act on important NETosis pathways through calcium ions [45]. In the present study, stimulation with ELF-PEMF did not affect Ca^2+^-influx in neutrophils (either with nor without additional PMA or LPS stimulation), suggesting that the exposure does not change the function of neutrophils through Ca^2+^ signaling either. Furthermore, from our results, the 16 Hz ELF-PEMF is not expected to modulate the induction of neutrophil activation and anti-inflammatory functions, which are normally modulated via Ca^2+^ signals [46].

In osteoprogenitor cells, increased Ca^2+^-influx of osteoprogenitor cells upon ELF-PEMF was regulated by the mechanosensitive Ca^2+^ ion channel Piezo1 [11]. Despite a previous report that neutrophils do not express *Piezo1* [47], we found robust *Piezo1* expression in neutrophils. However, blocking Piezo1 with the inhibitor Dooku1 did not show any changes in neutrophil Ca^2+^ signaling in response to ELF-PEMF, further proving ELF-PEMF exposure at 16 Hz frequency does not affect Ca^2+^ signaling in neutrophils. In neutrophils, G-protein coupled receptors (GPCRs) have been identified as mechanosensors [48], which can also induce Ca^2+^ influx by activating phosphoinositide 3-kinases and sphingosine kinase pathways [46]. Theoretically, GPCRs could be another candidate for the induction of Ca^2+^-influx into ELF-PEMF-exposed neutrophils. However, in the present study, 16 Hz ELF-PEMF did not affect Ca^2+^-influx and further mechanisms were not investigated.

To be more inclusive about the neutrophil response in clinical settings, LPS as a natural stimulus, as well as H_2_O_2_ and CI that use different pathways for activation than PMA [49], were further used to stimulate the neutrophils for NET release. To ensure the safety of the 16 Hz ELF-PEMF, it should not induce NET release with stimuli other than PMA. Neutrophils did not show enhanced NET release after stimulation with the above-mentioned compounds and ELF-PEMF exposure. Instead, after LPS and H_2_O_2_ stimulation, total DNA release was reduced. No modulation of NET formation at all was seen with a combination of CI and the exposure to the 16 Hz ELF-PEMF.

One could argue that reduced NET formation in response to LPS may cut down the anti-bacterial defense, but this seems to only be the case with a complete absence of NETs [50]. Therefore, we suggest that important pathways and mechanisms for neutrophil activation and neutrophil defense mechanisms are not negatively affected by 16 Hz ELF-PEMF exposure. This supports the safety of ELF-PEMF usage in clinical settings [51,52], where diverse complications with different molecular mechanisms can occur.

Various studies showed that different cell types can respond differently to particular ELF-PEMF fields. For instance, macrophages can show opposite responses to different fields, or they do not respond to fields that osteoblasts or osteoclasts respond to [22]. Moreover, so-called “frequency windows” were suggested for different types of cells [53]. In our study, none of the tested fields increased NET release, and several fields were even shown to slightly reduce NET formation. However, as seen before [11], different parameters of exposure (timing, several repeated exposures) could further influence the behavior of neutrophils. For clinical use directly after fracture, further exposure patterns might be worth investigating, especially in the highly dynamic immune response directly after an injury.

At the molecular level, the MAPK/ERK signaling cascade is known to regulate the cell response to ELF-PEMF exposure. One of our previous studies showed that the ERK1/2 pathway is strongly activated in primary human osteoblasts upon repeated ELF-PEMF exposure and is crucial for osteoblast function increase through ELF-PEMF exposure [52]. In addition, Poh et al., suggested that ELF-PEMF exposure prevented human adipose-derived mesenchymal stromal cell apoptosis by increasing levels of phosphorylated ERK1/2 and Akt [54]. Some studies demonstrate the activation of the ERK1/2 and MAPK cascade in planaria and mice upon ELF-PEMF exposure [55,56]. However, in neutrophils, the activation of MAPKs is associated with the induction of NETs formation [57]. In the present study, no differences in the protein levels of MAPK/ERK cascade were seen upon ELF-PEMF exposure. This is in line with the results, showing no change in ROS production, as it has been suggested that ROS generation by NADPH oxidase activates the ERK and p38 MAPK pathway [58]. Thus, our findings suggest that a single incidence of 16 Hz ELF-PEMF exposure does not activate the MAPK/ERK cascade in neutrophils and is not a prerequisite for NET formation. Apart from the bone fracture applications, ELF-PEMF has been studied for general wound-healing applications. It has been shown that EMF can help to balance free radicals and antioxidants by regulating inflammation in traumatized tissues [59]. ELF-PEMF exposure improved epidermal wound-healing in the inflammatory phase by inducing keratinocyte proliferation and modulating inflammatory response mediators [60]. Furthermore, it was seen that ELF-EMF can accelerate the transition from the inflammatory to the proliferative phase of wound-healing. ELF-PEMF exposure also affects the pathological matrix remodeling [61], cell migration, proliferation [62], angiogenesis [63], and expression of growth factors [64]. Collagen synthesis was increased [65], and skin ulcer-healing was promoted via ELF-EMF exposure in mice models [66]. These studies show that ELF-PEMF exposure has effects throughout all phases of healing processes. This study focused on effects relevant for the immediate use of ELF-PEMF after fracture and during the inflammatory phase of healing, and a similar reaction to ELF-PEMF exposure would be expected in neutrophils in the hemostatic phase of wound-healing. However, the in-depth investigation of the whole healing process, especially in patients with pro-inflammatory conditions, is needed in order to deem ELF-PEMF exposure safe for neutrophil function in different cases.

In light of our results, this project was able to show in terms of various aspects that 16 Hz and 7 min of ELF-PEMF exposure has no negative influence on neutrophil activation and NET release. Thus, the technology here described can be safely used immediately after a fracture without risking NET release, which is known to be detrimental to the healing process [30], and without the modulation of Ca^2+^ influx or ROS formation. It could even have a positive effect on healing through a slight reduction in NET formation.

## 4. Materials and Methods

### 4.1. Neutrophil Isolation

Venous blood was freshly collected in EDTA-tubes (S-Monovette 9 mL, Sarstedt, Germany). Neutrophil isolation was performed as previously described [53]. A total of 6 mL blood was layered on 6 mL of Lympholyte poly-cell separation medium (Cedarlane, Burlington, ON, Canada), followed by centrifugation at 500× *g* and 40 min at room temperature, without break. The plasma and PBMC layers were discarded, and the PMN layer was carefully collected into a fresh 15 mL tube. Cells were washed twice with 12 mL PBS and centrifuged at 400× *g* for 10 min at room temperature, at acceleration 5 and deceleration 4. The neutrophil pellet was resuspended in RPMI medium without phenol red (Sigma-Aldrich, Darmstadt, Germany). Cells were counted using the Trypan Blue exclusion method and a Neubauer counting chamber, omitting erythrocytes from the count.

### 4.2. ELF-PEMF Exposure

The ELF-PEMF devices (Somagen^®^, Sachtleben GmbH, Hamburg, Germany) were certified according to European law until May 2023 (CE 0482, compliant with EN ISO 13485:2016 [67] + EN ISO 14971:2012 [68]). These medical devices create an AC magnetic field and yield an inhomogeneous DC-field through the distortion of the local earth magnetic DC field, via the applicators [11]. In this study, mainly a 16 Hz fundamental frequency at an intensity of 6–282 µT was used, unless indicated otherwise. The exposure was carried out once by sending pulse bursts for 7 min. All tested fields (frequency in Hz: 10, 16, 20.6, 22.8, 23.8, 26, 28.8, 33, 34.8, 38.8, 40.8, 51.8, 71.8, 75.6, 90.6; results can be found in Appendix A) had the same pulse shape and burst pattern.

### 4.3. ROS Measurement

Total reactive oxygen species were measured using a 2′,7′-dichlorofluorescein-diacetate (DCFH-DA) assay. Briefly, 1 × 10^6^ neutrophils/mL were incubated in 10 μM DCFH-DA for 30 min at 37 °C. Cells were washed with PBS twice, centrifugation conditions were 400× *g*, 10 min, acceleration 5, and deceleration 4. Then, cells were exposed to the ELF-PEMF. Cells were stimulated with PMA or LPS as described previously and directly distributed into a 96-well plate. The measurement was carried out using the ClarioStar plus Microplate Reader (BMG Labtech, Ortenberg, Germany) for 2 h at 37 °C at Ex 485 nm/Em 520 nm. For analysis, values were normalized to untreated control cells at the 0 min measurement time point.

### 4.4. Ca^2+^ Measurement

Intracellular calcium was measured with Fura-2-AM (Chemodex, St. Gallen, Switzerland). Isolated neutrophils were incubated with 2 µM Fura-2-AM for 1 h at 37 °C in RPMI without phenol red. After washing the cells twice with HBSS (Sigma-Aldrich, Darmstadt, Germany), cells were incubated with 1 mM probenecid for 20 min at 37 °C to prevent the leakage of intracellular Fura-2 and then washed again with HBSS. For measurement, cells were resuspended in RPMI without phenol red with 1 mM additional Ca^2+^. For experiments with the inhibition of Piezo1, Dooku-1 (Sigma-Aldrich) was added at this time point in a concentration of 20 µM. ELF-PEMF exposure was performed as described above, followed by the direct addition of PMA or LPS. After stimulation, cells were directly transferred to a black 96-well plate with a clear bottom and measured in the ClarioStar plate reader (BMG Labtech) at two different wavelengths: Ca^2+^-loaded Fura-2 Ex 335 nm/Em 510 nm and Ca^2+^-free Fura-2 Ex 380 nm/Em 510 nm. The measurement was carried out for 2 h (every 2 min for the first 15 min, then every 5 min). For analysis, the ratio of Ca^2+^-loaded and Ca^2+^-free Fura-2 was calculated and normalized to EGTA-treated cells. The area under curve was calculated for the time frame of 10–70 min (peak time for PMA stimulation).

### 4.5. RNA Isolation and cDNA Synthesis

Total RNA isolation was performed via phenol-chloroform extraction. TriFast was added to the cells and incubated for 5 min at room temperature. A total of 100 μL chloroform per 500 µL Trifast was added (Carl Roth, Karlsruhe, Germany), incubated for 10 min in RT, and centrifuged at 14,000× *g* for 10 min at 4 °C. The upper aqueous phase, which contains the RNA, was collected carefully into a new tube containing 250 μL isopropanol (Honeywell, Charlotte, NC, USA). After incubation overnight at −20 °C, centrifugation with the same conditions as before was performed. The supernatant was discarded, and the RNA pellet was washed twice with 70% ethanol, followed by centrifugation. After discarding the supernatant, the tubes were left open to evaporate the remaining ethanol residues. The pellet was resuspended with DEPC H_2_O. The cDNA was synthesized using the First Strand cDNA Synthesis Kit (Thermo Fisher Scientific, Sindelfingen, Germany), according to the manufacturer’s instructions.

### 4.6. RT-PCR

Conventional polymerase chain reaction (RT-PCR) was used to observe the gene expressions of genes of interest. Red HS Taq (Biozym, Oldendorf, Germany) was used as the RT-PCR mastermix, according to the manufacturer’s instructions. Primer sequence and RT-PCR conditions are listed in Table 1.

### 4.7. Sytox Green Assay

Isolated neutrophils were prepared at 2 × 10^5^ cells/mL in RPMI without phenol red (Sigma-Aldrich, Darmstadt, Germany). Sytox Green (Thermo Fisher Scientific, Waltham, MA, USA) was added in a concentration of 1 μM. After exposure to ELF-PEMF, cells were immediately stimulated with 100 nM PMA, 25 µg/mL LPS, or 0.003% H_2_O_2_. For the normalization of the extracellular DNA, neutrophils were lysed with 1% Triton X-100 solution. The cells were incubated at 37 °C with 5% CO_2_ and the fluorescent intensity was measured at Ex 485 nm/Em 520 nm with a microplate reader every 30 min (FluoStar Omega, BMG Labtech, Ortenberg, Germany). Imaging was performed using a fluorescent microscope (EVOS FL, Life Technologies, Darmstadt, Germany) after 3 h of incubation.

### 4.8. Immunofluorescence

ELF-PEMF-exposed neutrophils suspension (150 μL, 3 × 10^5^ cells/mL) were seeded in sterile poly-L-lysine-coated chamber slides. After 4 h of incubation, neutrophils were fixed with 4% formaldehyde for 30 min and incubated with 1% Triton-X-100 for 5 min at room temperature. Fixed neutrophils were gently washed 3 times with PBS, incubated with 5% BSA solution for 1 h, and then incubated with the primary antibody overnight at 4 °C. Anti-MPO antibody (1:200 *v*/*v*, Cat. #sc-52707, Santa Cruz Biotechnology, Heidelberg, Germany) was used as the primary antibody. After the overnight incubation, neutrophils were gently washed 3 times with PBS and incubated with Alexa Fluor 488 goat anti-mouse IgG (1:1000 *v*/*v*, Cat. #A10667, Invitrogen, Heidelberg, Germany) for 2 h at room temperature. After gently washing 3 times, neutrophils were incubated with the 2 µg/mL Hoechst 33342 for 20 min to visualize the DNA. A fluorescence microscope was used to capture the images (EVOS FL, Life Technologies, Darmstadt, Germany). The ratio of NETosed cells was obtained by dividing the number of MPO^+^ cells by the number of Hoechst 33342^+^ cells. The analysis was performed as previously described [69].

### 4.9. Western Blot

For collection of protein, neutrophils were diluted to 1 × 10^6^ cells/mL and incubated with 100 nM PMA or 25 µg/mL LPS for 1.5 h. Neutrophils were collected through centrifugation (10 min, 400× *g*, *w*/*o* break, Acceleration = 5, Deceleration = 4) and lysed in ice-cold RIPA buffer (50 mM TRIS, 250 mM NaCl, 2% NP40, 2.5 mM EDTA, 0.1% SDS, 0.5% DOC, and protease/phosphatase inhibitors: 1 μg/mL pepstatin, 5 μg/mL leupeptin, 1 mM PMSF, 5 mM NaF, and 1 mM Na_3_VO_4_). The lysate was then centrifuged (10 min, 13,000× *g*, 4 °C) to remove cell debris. The protein concentration was measured via micro-Lowry assay [26]. The lysate was mixed with 5 × loading buffer at the ratio of 4:1 (*v*/*v*) and boiled at 95 °C for 10 min. A total of 35 μg protein were separated via SDS-PAGE (10% acrylamide-bisacrylamide gels, 100 V, 180 min) and subsequently transferred to nitrocellulose membranes (100 mA, 180 min). Protein separation and transfer were examined using Ponceau staining. The membrane was incubated with 5% BSA for 1 h at room temperature to block the unspecific binding sites. Membranes were subsequently incubated with primary antibodies at 4 °C overnight. Anti-cit-H3 antibody (1:790 *v*/*v*, Cat. #ab5103, Abcam, Heidelberg, Germany); Anti-phospho-Akt1/2/3 antibody (1:200 *v*/*v*, Cat. #sc-271966, Santa Cruz Biotech, Heidelberg, Germany); Anti-phospho-p38 antibody (1:1000 *v*/*v*, Cat. #4511, Cell Signaling, Heidelberg, Germany); Anti-phospho-ERK 1/2 antibody (1:1000 *v*/*v*, Cat. #4370, Cell Signaling, Heidelberg, Germany); and Anti-HPRT antibody (1:1000 *v*/*v*, Cat. #sc-376938, Santa Cruz Biotech, Heidelberg, Germany) were used in this study. After the overnight incubation, membranes were washed with TBS-T (25 mM Tris, 137 mM NaCl, 2.7 mM KCl, 0.05% Tween-20, pH = 7.4) three times and incubated with the corresponding HRP-labeled secondary antibodies (1:10,000 *v*/*v*; anti-mouse: Cell Signaling Technology #7076, anti-rabbit: Santa Cruz Biotechnology #2004) for 2 h at room temperature. Membranes were then covered with enhanced chemiluminescent substrate solution (ECL, 1.25 mM luminol, 0.2 mM p-coumaric acid, 0.03% H_2_O_2_ in 100 mM TRIS, pH = 8.5). A CCD camera was used to detect the chemiluminescent signals on the membrane (INTAS Chemocam, Göttingen, Germany). Signal intensities were determined using the ImageJ software and normalized to the HPRT signal, which was used as loading control.

### 4.10. Statistical Analysis

Data are shown as box plots with median, interquartile range, and 95% confidence interval (Tukey’s modification) where outliers are indicated as dots if not otherwise indicated. Data were analyzed via two-way ANOVA followed by Tukey’s multiple comparison test or non-parametric Kruskal–Wallis analysis using the GraphPad Prism software Version 8 (El Camino Real, CA, USA). *p* < 0.05 was considered statistically significant. The information on the statistical test is given in the figure legends. N indicates the number of donors; n indicates the number of technical replicates for each donor.

## Figures and Tables

**Figure 1 ijms-24-14629-f001:**
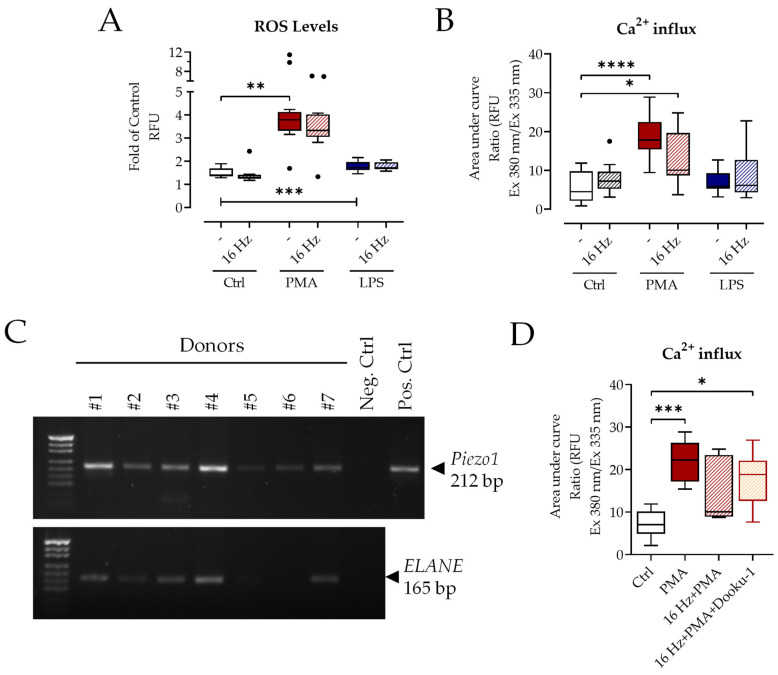
Exposure to 16 Hz ELF-PEMF does not change ROS formation or Ca^2+^-influx. Neutrophils were stimulated through ±100 nM PMA or ±25 µg/mL LPS after ±7 min exposure to 16 Hz ELF-PEMF. (**A**) Amount of total ROS formed was determined via DCFH-DA assay, N = 5, n = 3. (**B**) Ca^2+^-influx was quantified using Fura-2-AM measurement, N = 5, n = 3. (**C**) Results of *Piezo1* and *ELANE* PCR for neutrophils from seven different donors. (**D**) Ca^2+^-influx was quantified via Fura-2-AM measurement ± Piezo1 inhibitor Dooku-1 (20 µM), N = 5, n = 3. * *p* < 0.05, ** *p* < 0.01, *** *p* < 0.001, **** *p* < 0.0001, as determined via two-way ANOVA (**A**,**B**) or non-parametric Kruskal–Wallis test (**D**). Dots in the box plots indicate outliers. White bars: no additional stimulation, red bars: PMA stimulation, blue bars: LPS stimulation, shading indicates exposure to 16 Hz ELF-PEMF, orange indicates addition of Piezo-1 inhibitor.

**Figure 2 ijms-24-14629-f002:**
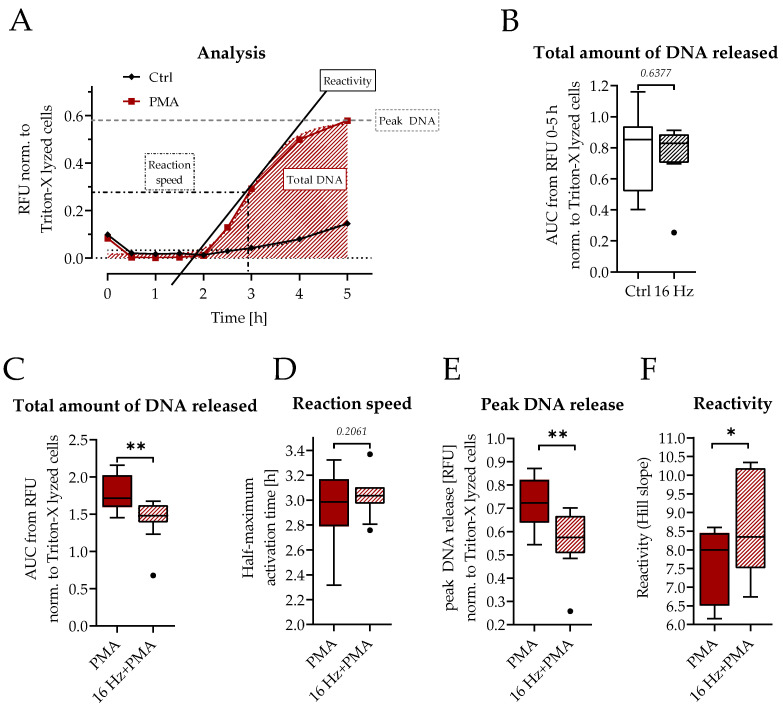
The 16 Hz EMF does not induce NET formation. (**A**) Time course of the Sytox Green assay of Ctrl- and PMA-stimulated neutrophils (no ELF-PEMF exposure), different analysis parameters are highlighted in the graph, shading indicates the area under the curve which represents the total amount of released DNA. (**B**) Total amount of DNA released obtained via the Sytox Green assay, ±7 min exposure of 16 Hz ELF-PEMF. N = 10, n = 3. (**C**–**F**) Different analysis parameters calculated from the Sytox Green assay with the addition of 100 nM PMA ±7 min of 16 Hz ELF-PEMF. N = 10, n = 3. (**C**) Total amount of released DNA. (**D**) Reaction speed. (**E**) Peak DNA release. (**F**) Reactivity. Statistical analysis was carried out using Wilcoxon’s test with * *p* < 0.05, ** *p* < 0.01. Dots in the box plots indicate outliers.

**Figure 3 ijms-24-14629-f003:**
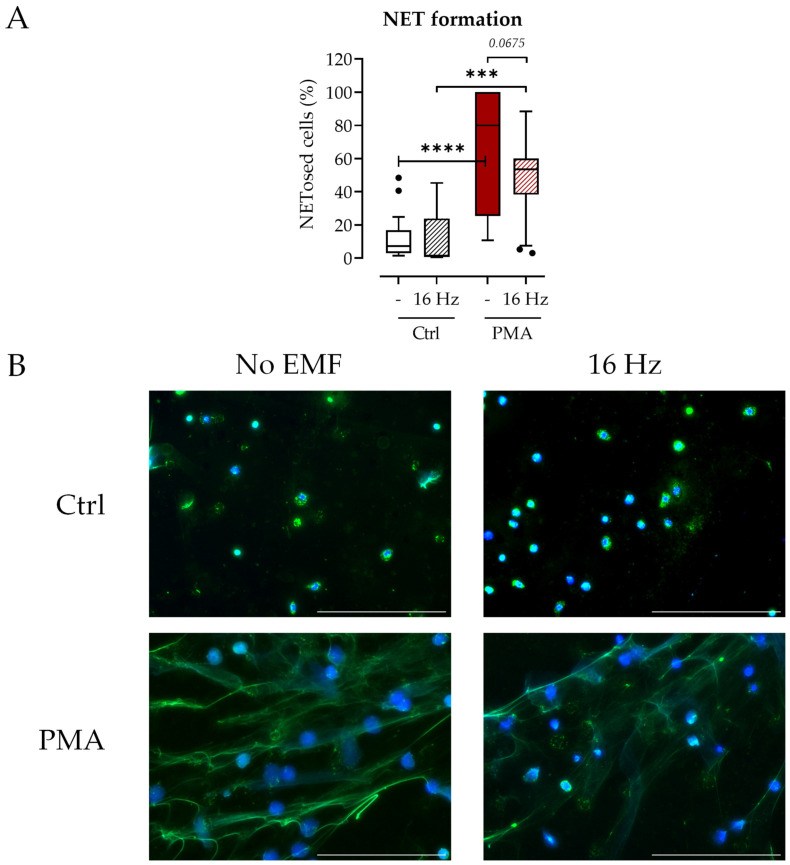
The 16 Hz ELF-PEMF exposure does not induce NET formation. Neutrophils were stimulated ±100 nM PMA after ±7 min exposure to 16 Hz ELF-PEMF. After 240 min, immunofluorescent staining for NETs was performed. (**A**) The number of NETosed cells was quantified from immunofluorescence images using the ImageJ software (Version 1.53), shading indicates exposure to 16 Hz ELF-PEMF, N = 3, n = 5. (**B**) Exemplary images of immunofluorescent stainings: Blue: DNA (Hoechst 33342); green: myeloperoxidase. Scale bar: 200 µm. *** *p* < 0.001 and **** *p* < 0.0001 as determined via two-way ANOVA. Dots in the box plots indicate outliers.

**Figure 4 ijms-24-14629-f004:**
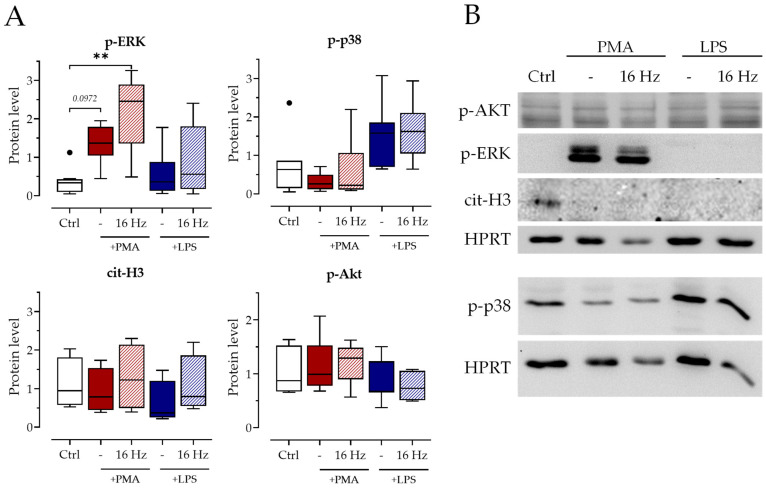
The 16 Hz ELF-PEMF exposure does not change MAPK activation. Neutrophils were stimulated ±100 nM PMA or ±25 µg/mL LPS after ±7 min exposure to 16 Hz ELF-PEMF, shading indicates exposure to 16 Hz ELF-PEMF. (**A**) Protein levels of indicated proteins determined via Western blot after 1.5 h of stimulation. (**B**) Exemplary blot images. N = 4, n = 2. ** *p* < 0.01, as determined by non-parametric Kruskal–Wallis test. Dots in the box plots indicate outliers.

**Figure 5 ijms-24-14629-f005:**
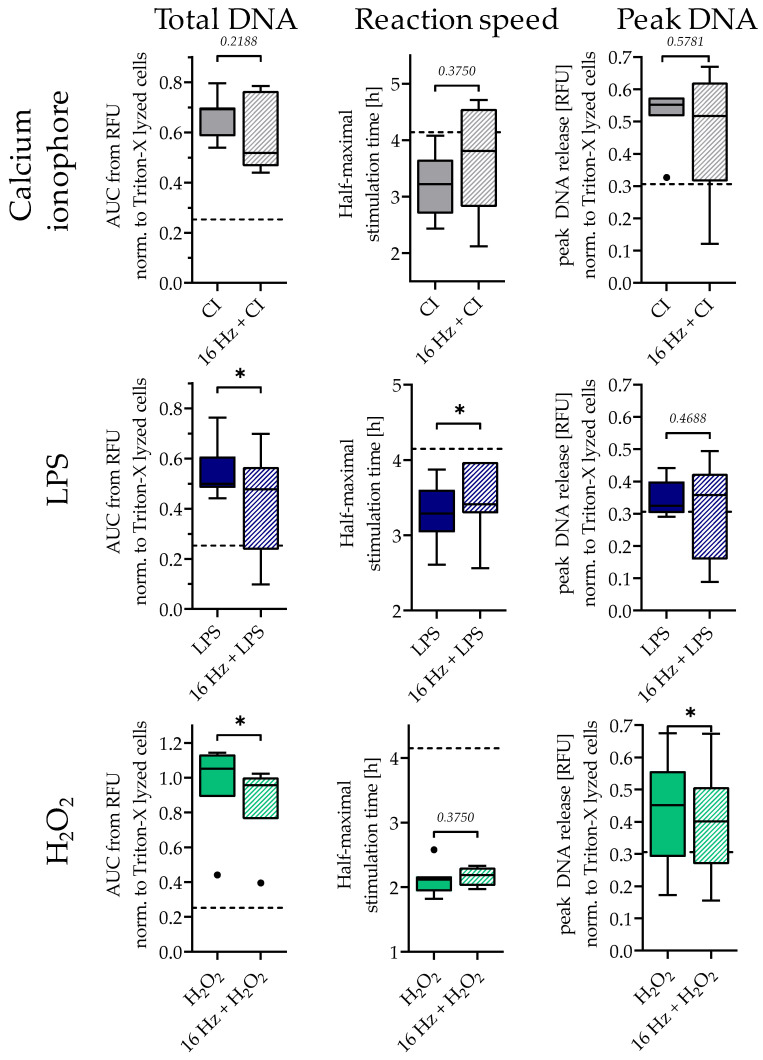
The 16 Hz ELF-PEMF exposure had a similar effect on NET formation to other stimulants. Total amount of DNA, activation time, and peak DNA release obtained from the Sytox Green assay. Neutrophils were stimulated using ±4 µM CI, ±25 µg/mL LPS, or ±0.003% H_2_O_2_ after ±7 min exposure of 16 Hz ELF-PEMF. N = 7, n = 4. Dashed line indicates control group without ELF-PEMF exposure. * *p* < 0.05, as determined using the Mann–Whitney U test. Dots in the box plots indicate outliers.

**Table 1 ijms-24-14629-t001:** List of PCR primers used in this study.

Primer	GenBank Accession	Forward Primer (5′-3′)	Reverse Primer (5′-3′)	Amplicon Size (bp)	T_a_ (°C)	Cycles
*Piezo1*	NM_001142864.4	ACCAACCTCATCAGCGACTT	AACAGGTATCGGAAGACGGC	212	56	40
*ELANE*	NM_001972.3	GCGTGGCGAATGTAAACGTC	ACCCGTTGAGCTGGAGAATC	165	58	40

## Data Availability

Datasets used are available from the corresponding author on reasonable request.

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
