# Peer review of "NET Formation Was Reduced via Exposure to Extremely Low-Frequency Pulsed Electromagnetic Fields"

_ijms, 2023, doi:10.3390/ijms241914629_

Round 1

Reviewer 1 Report

Thank you for submitting your study and for your dedication to the subject.

This manuscript covers an exciting and increasing topic worthy of investigation.

It has a good infrastructure, and the English language is fair.

However, the important problems should be revised.

This study mainly showed that the 16 Hz ELF-PEMF is safe and has no negative influence on NET. Additionally, this study showed the 16 Hz ELF-PEMF slightly reduced the NET formation. However, this study did not show that how the 16 Hz ELF-PEMF help the fracture healing. Slight reduction of NET formation cannot be explained to be helpful in fracture healing. This study should show that the 16 Hz ELF-PEMF helps fracture healing.

Minor editing of English language required

Reviewer 2 Report

Review of NET formation was reduced by exposure to extremely low frequency pulsed electromagnetic fields

In this study, the impact of a 16 Hz, extremely low-frequency pulsed electromagnetic field (ELF-PEMF) on fracture healing and neutrophil activation was investigated. ELF-PEMF did not induce harmful effects on neutrophils or exacerbate neutrophil extracellular trap (NET) formation, a process that can hinder healing. Instead, ELF-PEMF reduced DNA release during NET formation and demonstrated frequency-dependent effects. The findings suggest that ELF-PEMF, when applied for 7 minutes at 16 Hz, is safe for use immediately after a fracture, potentially benefiting healing by mitigating NET formation. Further research may optimize its application and explore broader clinical implications.

The subject of the study is important and the results are interesting. Before any positive remark, I have some comments and concern that needs to be addressed first.

Comments for authors

Comment 1: In evaluating this manuscript, it becomes evident that the introduction section lacks the necessary depth to fully convey the field's significance. To enhance the manuscript's introductory aspect, it is crucial to elucidate how electromagnetic field interacts with biological systems and to provide an up-to-date overview of the mechanisms involved. The incorporation of recent articles would serve as valuable resources for fortifying the background information and addressing this gap effectively.

Article: Microwave Radiation and the Brain: Mechanisms, Current Status, and Future Prospects. International Journal of Molecular Sciences vol. 23 (2022). [https://doi.org/10.3390/ijms23169288].

Comment 2.
What are the underlying molecular mechanisms by which ELF-PEMF exposure induces reactive oxygen species (ROS) and Ca2+ influx in bone cells, and how do these mechanisms differ from neutrophils?

Comment 3. Can the reduction in DNA release observed in neutrophils exposed to ELF-PEMF be attributed to alterations in DNA degradation pathways or nuclear envelope integrity?

Comment 4. Can the mechanisms described in this study be applied to other areas of regenerative medicine or tissue repair beyond fracture healing, and if so, what are the potential applications? Authors are encouraged to explain in the discussion section.

Comment 5. As authors mentioned in lines 312 – 313, “All tested fields (Frequency in Hz: 10, 16, 20.6, 22.8, 23.8, 26, 28.8, 33, 34.8, 38.8, 40.8, 51.8, 71.8, 75.6, 90.6) had the same pulse pattern.” Do authors have tested all those frequencies? I am unable to understand the intent of the authors in this statement. How does the pulse pattern remain the same if the frequency is different? Also, the cellular effects strongly depend on the frequency, so whether authors want to say these frequencies induce similar cellular effects.

Comment 6. Is there any specific reason to choose the particular 16 Hz frequency for investigations? This needs to be described in the manuscript.

Comment 7. The role of ROS in the present study needs to be discussed extensively to induce cellular effects in the discussion section.

Comment 8. The paper contains errors and typos that make it difficult to understand and distort its intended meaning. I encourage authors to reread carefully and fix any grammatical errors.

The paper contains errors and typos that make it difficult to understand and distort its intended meaning. I encourage authors to reread carefully and fix any grammatical errors.

Round 2

Reviewer 1 Report

Authors adressed the pointed issue well.

Reviewer 2 Report

The authors have addressed my comments in the revised version. I recommend accepting the article in its present form.